# Over-Expression of Two Different Isoforms of Cattle TUSC5 Showed Opposite Effects on Adipogenesis

**DOI:** 10.3390/genes13081444

**Published:** 2022-08-14

**Authors:** Han Xia, Fan Li, Lingwei Peng, Yuqin Du, Guohua Hua, Liguo Yang, Yang Zhou

**Affiliations:** 1Key Laboratory of Agricultural Animal Genetics, Breeding and Reproduction of Ministry of Education, Huazhong Agricultural University, Wuhan 430070, China; 2National Center for International Research on Animal Genetics, Breeding and Reproduction (NCIRAGBR), Huazhong Agricultural University, Wuhan 430070, China; 3Frontiers Science Center for Animal Breeding and Sustainable Production, Huazhong Agricultural University, Wuhan 430070, China

**Keywords:** TUSC5, alternative splicing, adipogenesis, RNA-seq

## Abstract

(1) Background: Adipogenesis is an important issue in human health and livestock meat quality that has received widespread attention and extensive study. However, alternative splicing events may generate multiple isoforms with different functions. This will lead to known knowledge being far more complex than before. (2) Methods: We studied the effects of two different TUSC5 isoforms (TUSC5A and TUSC5B) in cattle on adipogenesis by constructing over-expression cell models and RNA-sequencing methods. (3) Results: We discovered that over-expression of TUSC5A promotes the process of adipogenesis while over-expression of TUSC5B suppresses it. Eight important genes (*PPARG*, *ACC1*, *FASN*, *SCD1*, *LPL*, *FABP4*, *GPDH*, and *GLUT4*) during adipogenesis were significantly promoted (student’s *t*-test, *p* < 0.05) by TUSC5A and suppressed by TUSC5B both before and after cell differentiation. By performing a comprehensive analysis using a RNA-seq strategy, we found that both up-regulated differentially expressed genes (DEGs, |log2FoldChange| ≥ 1, *p* ≤ 0.05) of TUSC5A and down-regulated DEGs of TUSC5B were significantly enriched in the adipogenesis related GO terms, and the PPAR signaling pathway may play important role in those differences. (4) Conclusions: Our study proved that over-expression of two TUSC5 isoforms would regulate adipogenesis in the opposite direction. It is important to understand the function of the TUSC5 gene correctly.

## 1. Introduction

Adipogenesis is involved in important healthy issues that have drawn widespread attention in human studies [1]. Other than that, in livestock, it is a key factor affecting the economic value through altering fat mass distribution in muscle or not [2,3]. In cattle, the beef quality was evaluated according to the marbling score which can result in price differences of ten times the amount. To clarify, adipogenesis mechanisms are very important to understand in terms of the formation and distribution of adipose tissues. The TUSC5 plays a key role during adipogenesis [4]. It is robustly and specifically expressed in adipose tissue [5,6]. Further studies have proved that it is regulated by PPARG and modulates GLUT4 recycling to regulate insulin-mediated adipose tissue glucose uptake [7]. TUSC5 knockout mice exhibit impaired glucose disposal, and TUSC5 expression is predictive of glucose tolerance in obese individuals [4]. In cattle, we first identified two different isoforms of TUSC5 [8]. However, whether the two isoforms have the same functions during adipogenesis is unknown.

Previous studies showed that the alternative splicing event was common in mammalian genes and acted as a regulator of development and tissue identity [9,10]. Known mechanisms become far more complex than we thought when we start to consider different isoforms. It is reported that different isoforms of one gene may function differently or even have opposite functions. For example, alternative splicing converted STIM2 from an activator to an inhibitor of store-operated calcium channels [11]. A large number of genes have been detected with several isoforms that play roles during adipogenesis [12,13]. The alternative splicing of multiple genes including, TRA2B, BAG6, and MSH5 has been proven to be associated with human body mass index [14]. Alternatively, spliced MBNL1 isoforms were found to exhibit a differential influence on enhancing adipogenesis [15]. Furthermore, it was demonstrated that PPARG isoforms differentially regulated preadipocyte proliferation, apoptosis, and differentiation as a key gene regulating adipocyte differentiation [16,17]. The two isoforms of the cattle TUSC5 gene were found with 87 nucleotide differences that coded 29 amino acids before the 3rd exon and were named TUSC5A and TUSC5B. Our previous study showed that both the two isoforms are highly expressed in adipose tissue but show differential expression patterns during adipogenesis [18]. This illustrated that TUSC5A and TUSC5B might function differentially. It is essential to study the functions of the two TUSC5 isoforms to accurately understand the role of TUSC5 during adipogenesis.

Here, we created over-expression cell models by integrating coding sequences of TUSC5A and TUSC5B into the C3H10 cell line by lentivirus, separately. We proved that over-expression of TUSC5A and TUSC5B played opposite functions during adipogenesis by altering the expression of genes in multiple pathways for lipid accumulation. Our study detailed TUSC5 functions at different isoform levels and provided precise information for understanding TUSC5 gene functions during adipogenesis. This is necessary for further application of TUSC5 in cattle breeding to improve beef quality by regulating the distribution of adipose tissue.

## 2. Materials and Methods

### 2.1. Cell Culture

HEK 293T cells were cultured in high glucose medium (4.5 g/mL Glucose, 4.0 mM L-glutamine, cytiva, SH30243.01), DMEM containing 10% FBS supplemented with 1% penicillin-streptomycin at 37 °C and 5% CO_2_. The culture medium was changed every 24 h for lentivirus packaging. C3H10 T1/2 cells were cultured in high glucose (4.0 mM L-glutamine, EBSS, cytiva, SH30024.01) MEM supplemented with 10% FBS, 1% penicillin-streptomycin, and 1% NEAA (non-essential amino acid solution, Procell, PB180424) at 37 °C within a 5% CO_2_ environment. The culture solution was changed every 48 h.

### 2.2. Lentivirus Packaging

The coding sequences of TUSC5A and TUSC5B were cloned from cDNAs generated by reverse transcription of RNA extracted from cattle adipose tissue using the primers listed in Appendix A. For further study, we added sequences encoding 6His-tag at the end of the sequences of TUSC5A and TUSC5B in the primers during PCR amplification. Two restriction nucleases (BamH I and EcoR I) were applied to ligate the sequences of TUSC5A and TUSC5B to the coding region of the pCDH-CMV-MCS-EF1-copGFP-T2A-Puro vector using T4 DNA ligase (Thermo Scientific, Waltham, MA, USA). This vector contains the coding sequence for enhanced green fluorescent protein (cop-GFP). Subsequently, the ligated vector was transferred into *Escherichia coli* DH5α for further amplification and identification experiments. The successfully constructed pCDH-CMV-MCS-EF1-copGFP-T2A-Puro-TUSC5A/TUSC5B/control plasmid was transfected into the HEK 293T cell line for lentiviral packaging together with two other plasmids (pMD2.G and psPAX2). We placed HEK 293T cells in 6-well cell culture plates and transfected three plasmids (pCDH-CMV-MCS-EF1-copGFP-T2A-Puro-TUSC5A/TUSC5B/control:pMD2.G:psPAX2) in a ratio of 4:2:3 using transfection reagent (jetPRIME, Polyplus-transfection,114-15). The medium of HEK 293T was collected at 24 h, 48 h, and 72 h, respectively. Impurities such as cell debris were removed by low-speed centrifugation (4000× *g*, 5 min) and filtered through 0.22 μm pore size cellulose acetate filters.

### 2.3. Construction of Over-Expression Cell Models

To construct TUSC5A and TUSC5B over-expression cell models, C3H10 T1/2 cells were infected with lentivirus carrying TUSC5A or TUSC5B in the presence of 8 ug/mL Polybrene (Biosharp, Guangzhou, China) and screened with 4 ug/mL puromycin (Biosharp, Guangzhou, China) for 5–7 days. Subsequently, over-expression of TUSC5A and TUSC5B was confirmed using qPCR and Western blot. Information of primers used for qPCR are shown in Appendix A. 6His-tag antibody was used to detect overexpressed TUSC5A and TUSC5B proteins using Western blot. We used cop-GFP as a sorting condition to estimate the percentage of positive cells in the FL1 channel of the flow cytometer.

### 2.4. Assessment of Cell Proliferation

Cells of TUSC5A-C3H10, TUSC5B-C3H10, and control-C3H10 were first laid evenly on 6-well plates. After 48 h, cells were washed three times using phosphate buffer (Gibco™, America) and digested with 1 mL of trypsin (Gibco™, America) at 37 °C for 3 min each. The same numbers of the three kinds of cells were evenly spread out on a 24-well plate to ensure that they were at the same starting level. Cells were counted by an automated cell counter (Bio-Rad, Hercules, CA, USA) for each day.

### 2.5. Adipocyte Differentiation Assay

After cells of TUSC5A-C3H10, TUSC5B-C3H10, and control-C3H10 reached confluence in 6-well plates, a medium containing 1.0 μmol/L dexamethasone (DEX), 0.5 mmol/L 3-isobutyl-1-methylxanthine (IBMX), 1.0 μmol/L rosiglitazone, and 10 mg/L insulin were used to induce cells for 48 h. Then, fresh medium containing 10 mg/L insulin and 1.0 μmol/L rosiglitazone was used to maintain the differentiation of C3H10 cells. The culture solution was changed every 48 h.

### 2.6. Quantification of Triglyceride and Lipid Droplets

We used the GPO-PAP (Gliserolphosphat Oxidase Aminoantipyrine Peroxidase) method to determine the triglyceride levels [19]. Cells were washed using phosphate buffer (0.1 mol/L pH 7.4) before being collected after digestion with 0.25% trypsin (Gibco™, Waltham, MA, USA) and subsequently sonicated under ice water bath conditions. Absorbance reading was performed using a Spectrophotometer Biosystem A15 at a wavelength of 500 nm (Thermo Scientific, Waltham, MA, USA).

Oil Red O staining was performed using the Oil Red O staining kit (Solarbio, Beijing, China). Cells were washed 2–3 times using PBS (Gibco™, Waltham, MA, USA) before being fixed for 20–30 min. Cells were incubated with Oil Red O solution (0.33% *w*/*v* in 60% isopropanol) for 20 min at room temperature. The staining solution was removed and washed 4 times with double-distilled water. Images of Oil Red O-stained adipocytes were acquired using a Nikon imaging system (Nikon, TE2000-U, Tokyo, Japan) at 10× and 20× magnification for cells induced at the 4th, 6th, 8th, and 10th day. The lipid deposition stained by Oil Red O was evaluated by Image-Pro Plus 6.0 (Media Cybernetic, Rockville, MD, USA) software.

### 2.7. RNA Extraction and qPCR Assay

Total RNA was extracted from cultured C3H10 T1/2 cells using the FastPure Cell/Tissue Total RNA Isolation Kit V2 (Vazyme, Nanjing, China). Complementary DNA (cDNA) was reverse-transcribed from total RNA (1 µg) (Vazyme, China). Quantitative analysis of gene expression was performed by qPCR according to the SYBR Green I chimeric fluorescence kit (Vazyme, China) on the CFX Connect Real-Time PCR platform (BIO-RAD). Primers used in this study are shown in Appendix A. The qPCR conditions were as following: 5 min at 95 °C, 10 min at 95 °C, 40 cycles, 10 s at 95 °C, 10 s at 60 °C, and 15 s at 72 °C. The mRNA expression was normalized by comparison with the mouse cytoskeleton actin (ACTB, NC_000071.7).

### 2.8. Western Blot Assay

Cells were washed three times with pre-cooled PBS and analyzed using a high strength RIPA lysate (Servicebio, Wuhan, China) on ice after adding a protease inhibitor (Servicebio, Wuhan, China). Lysates were collected and centrifuged at 12,000× *g* for 5 min at 4 °C. The protein concentration was determined using the BCA protein concentration assay kit (Beyotime, Shanghai, China) and stored at −20 °C.

### 2.9. RNA Sequencing

Total RNA from TUSC5A-C3H10 cells, TUSC5A-C3H10 cells, and control-C3H10 cells were extracted using TRIzol (Invitrogen, Carlsbad, CA, USA) according to the manufacturer’s instructions. We measured the quantity and purity of RNA using a NanoDrop 8000 Spectrophotometer (NanoDrop Technologies, Wilmington, DE, USA) and Agilent 2100 Bioanalyzer System (Agilent Technologies, Santa Clara, CA, USA). Libraries were contracted and sequenced using the BGI T7 platform with paired-end (150 bp) reads.

NGSQCToolkit v2.3.3 was used to trim adapter sequences and low-quality reads. The clean reads were aligned on the reference genome (ARS-UCD1.2) along with annotated genes in the Ensembl website (https://www.ensembl.org/index.html (25 June 2021)) using the HISAT2 v2.1.0 with the default parameters [20]. The spliced reads were assembled to transcripts using the StringTie v1.3.3 software for each sample. Differentially expressed genes were measured using DESeq2 v3.15 [21]. Gene functional annotation analyses were applied using the online DAVID software (https://david.ncifcrf.gov/ (30 July 2021)) [22]. The Fisher’s exact test was used to measure gene enrichment in annotation terms. *p* values were corrected by FDR to search for significantly enriched terms.

## 3. Results

### 3.1. Over-Expression of TUSC5B but Not TUSC5A Inhabited C3H10 Cell Proliferation

To study the effect of the two different isoforms of cattle TUSC5 on adipogenesis, we constructed over-expressed cell models by transferring TUSC5A and TUSC5B into the genome of C3H10 cells by lentivirus, separately (Figure 1a). After multiple times of puromycin screening, we received highly purified positive cells (>98%, Appendix A) for C3H10 cells over-expressing TUSC5A and TUSC5B. Both the two TUSC5 isoforms were over-expressed thousands of times in C3H10 cells (Figure 1b,c). We first asked whether over-expression of two different TUSC5 isoforms would affect the proliferation of C3H10 cells. By plotting the growth-curve, we observed that TUSC5A-C3H10 cells (C3H10 cells over-expressing TUSC5A) and TUSC5B-C3H10 cells (C3H10 cells over-expressing TUSC5B) increased with similar speed before the 4th day (Figure 1d). However, after that, TUSC5B-C3H10 cells were with a significantly lower proliferation speed (student’s *t*-test, *p* < 0.05) than control-C3H10 cells and TUSC5B-C3H10 cells in the exponential growth stage (Figure 1d). We applied flow cytometry to check the cell statues after being plated for 48 h. Both the proportion of early and late stages showed slightly more apoptosis in TUSC5B-C3H10 cells than in control-C3H10 cells and TUS5A-C3H10 cells, which might be the reason for the lower proliferation speed of TUSC5B-C3H10 cells (Figure 1e and Appendix A).

### 3.2. TUSC5A Promoted While TUSC5B Inhibited Lipid Accumulation during Adipogenesis

We recorded the lipid accumulation statues of TUSC5A-C3H10 cells and TUSC5B-C3H10 cells during adipogenesis. On the 4th day of cell differentiation, we saw that lipid droplets in TUSC5A-C3H10 cells were more accumulated while much fewer lipid droplets were seen in TUSC5B-C3H10 cells (Figure 2a). This difference was maintained at the 6th, 8th, and 10th day of cell differentiation (Figure 2b). We further evaluated triglyceride levels, after cell induced, at 0 day, 2 day, 4 day, 6 day, 8 day, and 10 day. Before the induced differentiation, both triglyceride levels in TUSC5A-C3H10 cells and TUSC5B-C3H10 cells were higher than control-C3H10 cells (Figure 2c). However, after that, we observed that TUSC5A-C3H10 and TUSC5B-C3H10 cells showed the opposite lipogenic capacity compared to control-C3H10 cells (Figure 2c). The triglyceride level in TUSC5B-C3H10 cells became significantly lower (student’s *t*-test, *p* < 0.05) than that of control-C3H10 cells throughout all the stages of adipogenesis. At the same time, the triglyceride level in TUSC5A-C3H10 cells was kept higher than that of control-C3H10 cells during adipogenesis. Thus, we concluded that over-expression of TUSC5A significantly promoted lipid accumulation while it was significantly suppressed by over-expression of TUSC5B.

### 3.3. Expressions of Adipogenesis Important Genes Were Affected by TUSC5A and TUSC5B

We checked the expression of eight important genes (*PPARG*, *ACC1*, *FASN*, *SCD1*, *LPL*, *FABP4*, *GPDH*, and *GLUT4*) of adipogenesis before and after cell differentiation using qPCR and Western blot. The result showed that all those eight genes were significantly up-regulated (student’s *t*-test, *p* < 0.05) in the TUSC5A-C3H10 cells while they were significantly suppressed in TUSC5B-C3H10 cells compared to that of control-C3H10 cells (Figure 3a,b). After being induced for adipogenesis, the eight genes exhibited different expression patterns but with a consistent result that all of them were promoted in TUSC5A-C3H10 cells and suppressed in TUSC5B-C3H10 cells after the 4th day (Figure 3c). The expression of *PPARG*, *ACC1*, *FASN*, and *GPDH* were put off for at least two days while the expression of *FABP4*, *LPL*, *SCD1*, and *GLUT4* were suppressed throughout the 10 days in TUSC5B-C3H10 cells (Figure 3c).

### 3.4. Global Differences of Gene Expression between TUSC5A-C3H10 Cells and TUSC5B-C3H10 Cells Detected by RNA Sequencing

We performed RNA sequencing on the 4th day of differentiation of cells for TUSC5A-C3H10, TUSC5B-C3H10, and control-C3H10 with three replicates. Each sample generated an average of 50 million short reads. The PCA analysis showed that the three replicates for each cell type were clustered together, which proved the high quality of replicates from the same cell types (Figure 4a). The TUSC5A-C3H10, TUSC5B-C3H10, and control-C3H10 cells were separated by PC1. The control-C3H10 is, as expected, in the middle of the TUSC5A-C3H10 and TUSC5B-C3H10 (Figure 4a). This is consistent with the lipid droplet statues and proved that the two isoforms of TUSC5 were opposite in regulating adipogenesis at the global RNA expression level.

We analyzed differentially expressed genes (DEGs, |log2FoldChange| ≥ 1, *p* ≤ 0.05) between each pair of the three cell types. There were 531 DEGs between TUSC5A-C3H10 cells and control-C3H10 cells, while a much larger number of DEGs (3609) were detected in TUSC5B-C3H10 cells compared to control-C3H10 (Figure 4b). The DEG number (4401) between TUSC5A-C3H10 cells and TUSC5B-C3H10 cells was the largest, which included 72.32% DEGs of TUSC5A-C3H10 cells and 78.25% DEGs of TUSC5B-C3H10 cells compared to control-C3H10 cells (Figure 4b). The DEGs provided more evidence that TUSC5A-C3H10 cells and TUSC5B-C3H10 cells would lead to the opposite regulation of gene expression (Figure 4c). Moreover, the expression of all the eight genes in the RNA-seq was consistent with the previous qPCR results.

### 3.5. Functional Enrichment Analyses of DEGs

We applied Gene Ontology (GO) and KEGG pathway enrichment analyses for DEGs of each pair separately. Up-regulated DEGs of TUSC5A-C3H10 cells compared to control-C3H10 cells were significantly (FDR < 0.05) enriched in adipogenesis related GO terms including lipid metabolic process, fatty acid metabolic process, etc. (Figure 4d). Most of them were also significantly enriched (FDR < 0.05) in the down-regulated DEGs of TUSC5B-C3H10 cells compared to control-C3H10 cells (Figure 4d). It is to be noted that those adipogenesis-related GO terms became more significant in the GO enrichment result of DEGs between TUSC5A-C3H10 cells and TUSC5B-C3H10 cells (Figure 4d).

We collected pathways that were both significantly (FDR < 0.05) enriched for DEGs that were up-regulated in TUSC5A-C3H10 cells comparing to control-C3H10 cells and DEGs that were down-regulated in TUSC5B-C3H10 cells compared to control-C3H10 cells (Appendix A). As a result, the PPAR signaling pathway and other two metabolism-related pathways (fatty acid metabolism and metabolic pathways) met this criterion. Eight genes (*FABP4*, *FABP5*, *ACSL1*, *EHHADH*, *ADIPOQ*, *NR1H3*, *LPL*, and *SCD1*) in the PPAR signaling pathway were significantly up-regulated in TUSC5A-C3H10 cells while significantly down-regulated in TUSC5B-C3H10 cells.

## 4. Discussion

Here, we studied the effects of different TUSC5 isoforms on adipogenesis through over-expressing TUSC5A and TUSC5B in C3H10 cells, separately. After being induced for adipogenic differentiation, TUSC5A-C3H10 cells were significantly promoted while TUSC5B-C3H10 cells were significantly suppressed in the process of adipogenesis. This kind of phenomenon has been reported in multiple other genes [11,15,17]. Isoforms of one gene with different functions would change our knowledge of it [23]. This would finally affect its application using molecular breeding, genome selection, or gene editing techniques to regulate fat mass distribution in cattle. To date, a great number of isoforms have been identified through high-throughput sequencing and proved that most genes have alternative splicing [24,25]. It is becoming more important and urgent to clarify their potential different functions.

We evaluated the different effects on C3H10 cells by over-expression of TUSC5A and TUSC5B through performing multiple experiments at different levels. This includes qPCR, Western blot and, RNA-sequencing at the molecular level, and lipid staining, etc., at the phenotype level. All the results were consistent and proved that the over-expression of TUSC5A would promote while TUSC5B would suppress the process of adipogenesis. Thus, we received solid evidence for the different functions of the two TUSC5 isoforms.

It is noted that the lentivirus bring the sequences of TUSC5 isoforms and inserted them into the genome of C3H10 cells, which would express TUSC5 isoforms continuously [26]. Previous studies reported that the TUSC5 gene was activated by PPARG and was robustly expressed in the middle to late stages of adipogenesis [7]. In this study, the TUSC5 isoforms were over-expressed in the early stage of adipogenesis in C3H10 cells, which brought the expression peak forward and was out of the control of PPARG. The expression of PPARG, together with the other seven adipogenesis-related genes, was significantly suppressed in TUSC5B-C3H10 cells before and after cell-induced differentiation. This suggests that robust over-expression of TUSC5 isoforms in the early stage of adipogenesis would change the normal expression of those genes by feedback regulation of PPARG and other genes. At the same time, the PPAR signaling pathway was commonly enriched for DEGs of TUSC5A-C3H10 cells and TUSC5B-C3H10 cells compared to control-C3H10 cells. The PPAR signal pathway might be one of the most related reasons responsible for differences in adipogenesis between TUSC5A-C3H10 cells and TUSC5B-C3H10 cells. However, details of the mechanism still need to be clarified by further studies. Moreover, the in vivo experiment is necessary to prove the different effects of TUSC5A and TUSC5B will be maintained. This is an important step toward promoting the use of TUSC5 isoforms in the regulation of adipogenesis.

## 5. Conclusions

In this study, we proved that the two isoforms of cattle TUSC5 would have opposite effects on adipogenesis, which might be through regulating the PPAR signaling pathway. This will be important for corrective understanding of the TUSC5 function and promoting the application of TUSC5 in cattle breeding.

## Figures and Tables

**Figure 1 genes-13-01444-f001:**
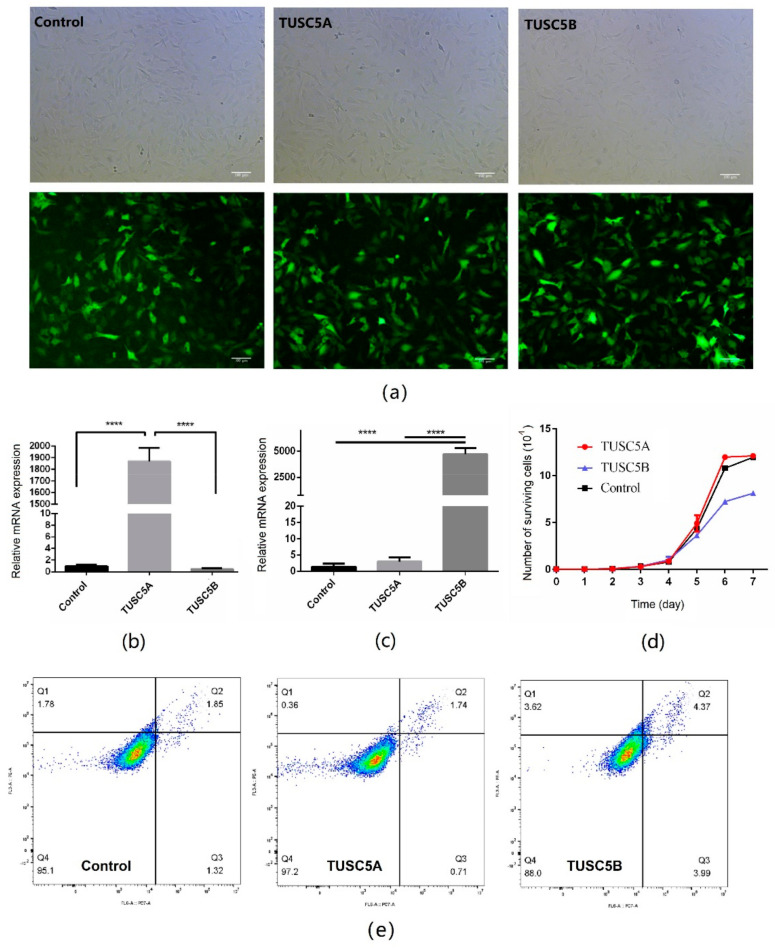
Over-expression cell models for different isoforms of TUSC5. (**a**) images for cell models of TUSC5A-C3H10, TUSC5B-C3H10, and control-C3H10. Green: cells that successfully transferred vectors by lentivirus and expressed cop-GFP protein; (**b**) relative TUSC5A mRNA expression level for cell models of TUSC5A-C3H10, TUSC5B-C3H10, and control-C3H10; (**c**) relative TUSC5B mRNA expression level for cell models of TUSC5A-C3H10, TUSC5B-C3H10, and control-C3H10; (**d**) growth-curves for cell models of TUSC5A-C3H10, TUSC5B-C3H10, and control-C3H10. Control: C3H10 cells were transferred with empty vector by lentivirus; TUSC5A: Control: C3H10 cells were transferred TUSC5A coding sequence by lentivirus; TUSC5B: Control: C3H10 cells were transferred TUSC5B coding sequence by lentivirus. Q1: shows necrotic cells, Q2: shows later period apoptotic cells, Q3: shows normal cells and Q4: shows early apoptotic cells. (**e**) cell statues after being plated for 48 h checked by flow cytometry. ****: *p* < 0.0001.

**Figure 2 genes-13-01444-f002:**
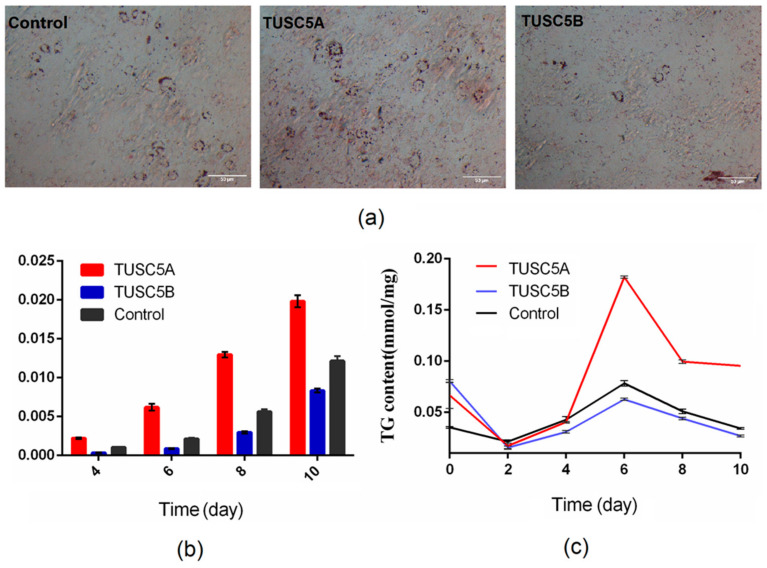
Quantification of triglyceride and fat droplets during adipogenesis. (**a**) images for Oil Red O staining at 4th day of adipogenesis; (**b**) lipid accumulation during adipogenesis on the 4th, 6th, 8th, and 10th day. The y axis indicates the ratio of area stained by red oil in the image; (**c**) variation of triglyceride levels during adipogenesis on the 0th, 2nd, 4th, 6th, 8th, and 10th day.

**Figure 3 genes-13-01444-f003:**
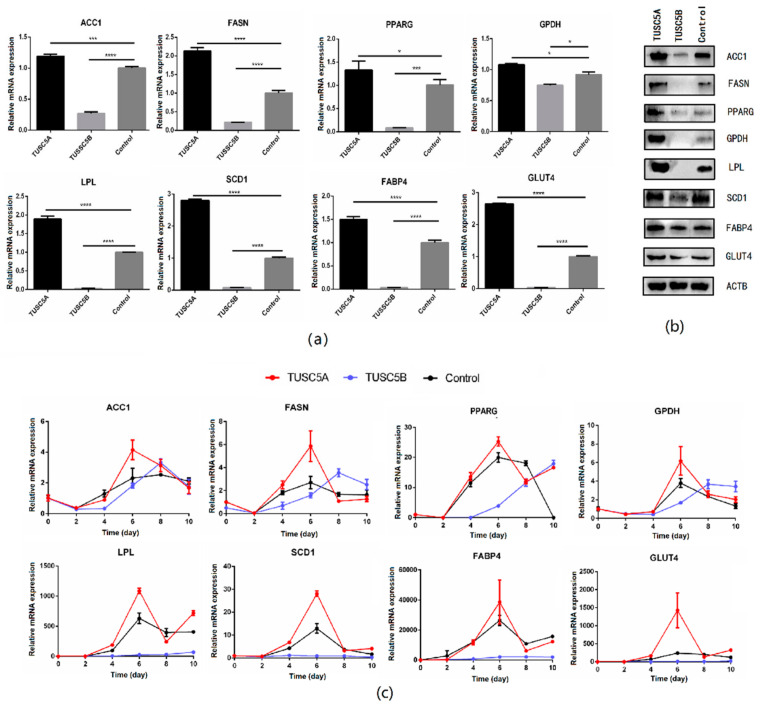
Quantification of gene expression before and after cell differentiation. (**a**) qPCR quantification of gene expression before cell differentiation; (**b**) Western blot quantification of gene expression before cell differentiation; (**c**) qPCR quantification of gene expression during adipogenesis. *: *p* < 0.05; ***: *p* < 0.001; ****: *p* < 0.0001.

**Figure 4 genes-13-01444-f004:**
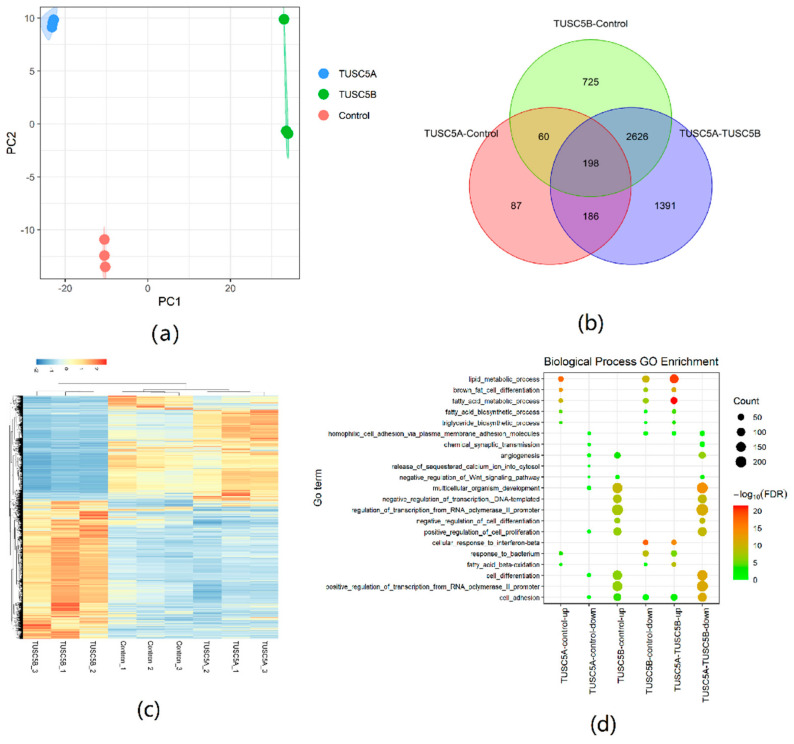
Differentially expressed gene analyses using RNA sequencing. (**a**) PCA analysis using a gene expression dataset; (**b**) Venn plot for DEG number; (**c**) Heatmap plot for DEG expression level; (**d**) DEG gene ontology analyses.

## Data Availability

The RNA sequencing datasets generated by this study were deposited in the NCBI SRA database with accession ID: PRJNA851185.

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
