# Peer review of "Over-Expression of Two Different Isoforms of Cattle TUSC5 Showed Opposite Effects on Adipogenesis"

_genes, 2022, doi:10.3390/genes13081444_

Round 1

Reviewer 1 Report

Dear authors, this is a sound manuscript and science was done accordingly. However, you have to improve language writting. Please, review

Other points:

1) Figure1: please, explain in the legend what does letter (e) means;

2) At the sentence from line 189 to 191: "Both the proportion of early and late stages showed slightly more apoptosis in  TUSC5B-C3H10 cells than control-C3H10 celsl and TUSC5B-C3H10 cells, which might be the reason for the lower proliferation speed of TUSC5B-C3H10 cells", I believe there is a confusion on the nomiantion of the cell lines. There is also some words that are mispelled

I suggest you go trought the text and look for other confisions in the nomination of the cell lines and mispelled words

3) Figure 2 (b): What does the Y axis means?

4) Figure 3: is it (a), (b) or (c), or is it (1), (2) and (3)?

5) Statistical methods: Which were the methods used to compare: 

5.1.) Differences in cell proliferation between treatments?

5.2) Differences in Adipocyte differentiation between treatments?

5.3) Differences in Quantification of triglyceride and lipid droplets between treatments?

5.4)  Differences in qPCR assay results between treatments?

Author Response

Dear authors, this is a sound manuscript and science was done accordingly. However, you have to improve language writting. Please, review

A: Done. Thank you!

Other points:

1) Figure1: please, explain in the legend what does letter (e) means;

A: Done. Thank you! We added the information in the legend as following: Control: C3H10 cells were transferred none sequence by lentivirus; TUSC5A: Control: C3H10 cells were transferred TUSC5A coding sequence by lentivirus; TUSC5B: Control: C3H10 cells were transferred TUSC5B coding sequence by lentivirus. Q1: shows necrotic cells, Q2: shows later period apoptotic cells, Q3: shows normal cells and the Q4: shows early apoptotic cells.

2) At the sentence from line 189 to 191: "Both the proportion of early and late stages showed slightly more apoptosis in  TUSC5B-C3H10 cells than control-C3H10 celsl and TUSC5B-C3H10 cells, which might be the reason for the lower proliferation speed of TUSC5B-C3H10 cells", I believe there is a confusion on the nomiantion of the cell lines. There is also some words that are mispelled

I suggest you go trought the text and look for other confisions in the nomination of the cell lines and mispelled words

A: Thank you very much. We corrected the misspelled word and error typing cell line names.

3) Figure 2 (b): What does the Y axis means?

A: The Y axis means the ratio of area stained by red oil in the image. We added it in the figure legend. Thank you.

4) Figure 3: is it (a), (b) or (c), or is it (1), (2) and (3)?

A: It is (a), (b) or (c). We have corrected it. Thank you!

5) Statistical methods: Which were thAA:e methods used to compare: 

5.1.) Differences in cell proliferation between treatments?

5.2) Differences in Adipocyte differentiation between treatments?

5.3) Differences in Quantification of triglyceride and lipid droplets between treatments?

5.4)  Differences in qPCR assay results between treatments?

A: Thank you! It’s Student’s t-test. We added it in the manuscript.

Reviewer 2 Report

The scientific manuscript is well written.
Part of the chapter Discussion (lines 284 to 291) is written in a form more appropriate for the chapter Results of this research. Therefore, I suggest that this paragraph be reformulated.

I suggest adding a sentence in the chapter Conclusion as a recommendation for further research on this topic.

Author Response

The scientific manuscript is well written.
Part of the chapter Discussion (lines 284 to 291) is written in a form more appropriate for the chapter Results of this research. Therefore, I suggest that this paragraph be reformulated.

A: Done. Thank you!Please see line 289-294.

I suggest adding a sentence in the chapter Conclusion as a recommendation for further research on this topic.

A: Done. Thank you!Please see line 317-320.
